Histopathological analysis of zebrafish after introduction of non-biodegradable polyelectrolyte microcapsules into the circulatory system

Borvinskaya Ekaterina borvinska@gmail.com 1
Gurkov Anton 1 2
Shchapova Ekaterina 1 2
Mutin Andrei 1
Timofeyev Maxim 1 2
1 Institute of Biology at Irkutsk State University , Irkutsk , Russia
2 Baikal Research Centre , Irkutsk , Russia
Nganvongpanit Korakot
Electronic publication date: 2021 May 5
Publication date: 2021
Volume: 9
Electronic Location ID: e11337
Received 2020 Nov 17; Accepted 2021 Apr 2
Copyright: ©2021 Borvinskaya et al.
Copyright year: 2021
Copyright holder: Borvinskaya et al.
License: This is an open access article distributed under the terms of the Creative Commons Attribution License, which permits unrestricted use, distribution, reproduction and adaptation in any medium and for any purpose provided that it is properly attributed. For attribution, the original author(s), title, publication source (PeerJ) and either DOI or URL of the article must be cited.
License URL: https://creativecommons.org/licenses/by/4.0/

Keywords: Polyelectrolyte microcapsules, Danio rerio, Histology, Implant response, Biocompatibility, Inflammation, Layer-by-layer encapsulation, Polymer science, In vivo sensing

Funding: Russian Science Foundation #20-64-47011 Petrozavodsk State University (Petrozavodsk, Russia) Irkutsk State University with the support of the Russian Foundation for Basic Research #19-34-90137 The histological analysis was supported by the Russian Science Foundation (#20-64-47011) for interdisciplinary project performed in association with the Petrozavodsk State University (Petrozavodsk, Russia). Keeping, care and experimental manipulations with animals were carried out in Irkutsk State University with the support of the Russian Foundation for Basic Research (#19-34-90137). There was no additional external funding received for this study. The funders had no role in study design, data collection and analysis, decision to publish, or preparation of the manuscript.

==============================
Polyelectrolyte microcapsules are among the most promising carriers of various sensing substances for their application inside the bloodstream of vertebrates. The long-term effects of biodegradable microcapsules in mammals are relatively well studied, but this is not the case for non-biodegradable microcapsules, which may be even more generally applicable for physiological measurements. In the current study, we introduced non-biodegradable polyelectrolyte microcapsules coated with polyethylene glycol (PMs-PEG) into the circulatory system of zebrafish to assess their long-term effects on fish internal organs with histopathologic analysis. Implantation of PMs-PEG was not associated with the formation of microclots or thrombi in thin capillaries; thus, the applied microcapsules had a low aggregation capacity. The progression of the immune response to the implant depended on the time and the abundance of microparticles in the tissues. We showed that inflammation originated from recognition and internalization of PMs-PEG by phagocytes. These microcapsule-filled immune cells have been found to migrate through the intestinal wall into the lumen, demonstrating a possible mechanism for partial microparticle elimination from fish. The observed tissue immune response to PMs-PEG was local, without a systemic effect on the fish morphology. The most pronounced chronic severe inflammatory reaction was observed near the injection site in renal parenchyma and within the abdominal cavity since PMs-PEG were administered with kidney injection. Blood clots and granulomatosis were noted at the injection site but were not found in the kidneys outside the injection site. Single microcapsules brought by blood into distal organs did not have a noticeable effect on the surrounding tissues. The severity of noted pathologies of the gills was insufficient to affect respiration. No statistically significant alterations in hepatic morphology were revealed after PMs-PEG introduction into fish body. Overall, our data demonstrate that despite they are immunogenic, non-biodegradable PMs-PEG have low potential to cause systemic effects if applied in the minimal amount necessary for detection of fluorescent signal from the microcapsules.

Introduction

Various blood parameters are usually considered reliable and comprehensive markers of the physiological state of any vertebrate organism (Tahir et al., 2018; Bakke et al., 2020; Huang et al., 2019). Most current approaches require blood extraction to analyze its parameters; however, continuous real-time signal acquisition directly from an animal bloodstream has long been a desirable goal in physiological research (Wang et al., 2017; Gray et al., 2018). A variety of recently developed molecular, nano- and microscale probes that may pass the information about blood parameters via optical, acoustic or magnetic signal (Rong et al., 2019). However, most of them require a carrier for implantation into the circulatory system to concentrate them to a restricted volume, minimize their possible influence on the organism, and temporarily hide them from the immune system (Shibata et al., 2010; Lee et al., 2018; Kanick et al., 2019).

Polyelectrolyte microcapsules (PMs) are one of the most promising candidates as such carriers. PMs, assembled by a layer-by-layer adsorption technique (Donath et al., 1998; Volodkin, Larionova & Sukhorukov, 2004; Borvinskaya et al., 2018b), are hollow, several microns in size, and have an elastic semipermeable wall, which mimics erythrocytes in the ability to disseminate across the circulatory system, even in the smallest blood capillaries, without considerable occlusions (She et al., 2012). Furthermore, PMs are easy to prepare, do not contain toxic components and can be covered by such coatings as polyethylene glycol (PEG) to delay their recognition by the immune system (Wattendorf et al., 2008).

Previously, we successfully used PMs for the delivery of the pH-sensitive molecular probe (SNARF-1) to gill capillaries of fish and performed local pH measurements in real time (Borvinskaya et al., 2017). Adapting other sensing substances for application inside PMs implanted into bloodstream of different vertebrates may create a wide palette of measuring tools for various physiological tasks, however this requires prior testing of how PMs affect the animal circulatory system.

PMs can be assembled from either biodegradable and non-biodegradable polymers, which begins to be important when they are eventually engulfed or surrounded by immune cells. In this case biodegradable PMs can be dissolved, while non-biodegradable ones stay intact inside the cells or in the intercellular spaces for a long time (De Geest et al., 2006; Borvinskaya et al., 2018a; Shchapova et al., 2019). In the context of long-term or repetitive physiological measurements in the bloodstream, biodegradable PMs are more convenient for carrying non-toxic probes sensitive to parameters that cannot be measured from intracellular space, such as blood pH. Oppositely, PMs assembled from non-biodegradable polymers may be of interest in combination with some toxic probes or to identify compounds that easily penetrate cellular membranes, such as oxygen. The amount of the toxic probes anchored inside biodegradable PMs is low, and they would barely affect the whole organism; however, their toxicity would be the most pronounced right at the spots of signal acquisition, which can indirectly influence the experimental results. Moreover, the application of non-biodegradable carriers reduces the toxicity of most microencapsulated probes to the toxicity of the capsules themself. Thus, non-biodegradable PMs may be considered as a more generally applicable tool for purposes of physiological sensing.

Several studies describing the distribution and influence of layer-by-layer assembled microcapsules after injection into the mouse bloodstream (Yi et al., 2014; Shao et al., 2015; Voronin et al., 2017; Navolokin et al., 2018; Sindeeva et al., 2020) showed no significant deleterious effects. However, as the main focus of research in this field is drug delivery, these studies utilized biodegradable microcapsules or included short-term experiments, while information on whole-organism long-term effects of systemic administration of non-biodegradable microcapsules is scarce. Previously, we had the first glimpse at the distribution and behavior of non-biodegradable fluorescent PMs one week after introduction into the circulatory system of adult zebrafish Danio rerio (Hamilton, 1822) and observed their concentration in capillary-rich organs (Borvinskaya et al., 2018a). Zebrafish was chosen as the model object due to its small size and applicability for studying the principles of vertebrate morphology (Ding et al., 2019; Tonelli et al., 2020). External examination of the fish organs revealed sporadic accumulations of microcapsules in the capillaries of the kidney, gills, liver, spleen, and brain, but it was unclear whether these aggregates were temporary or had the potential to clog the capillaries completely. Nevertheless, we observed no significant pathological patterns in the internal organs.

Here we aimed to perform a comprehensive and long-term study of the influence of non-biodegradable PMs implanted into the bloodstream on the internal organs of zebrafish with a focus on histopathological analysis to reveal possible generalized and chronic inflammation, clogging of small capillaries, or determine which organs may be the most affected by the implant. As the used microcapsules were coated with polyethylene glycol for increasing their biocompatibility, we refer to them as PMs-PEG.

Materials and Methods

Preparation of PMs-PEG

Fluorescent PMs-PEG were fabricated as described in detail previously (Borvinskaya et al., 2018b). Conjugate of the fluorescent dye fluorescein isothiocyanate with albumin (#FD20S; Sigma-Aldrich, USA; Ex 494 nm, Em 512 nm) was microencapsulated in order to contrast PMs-PEG in fish organs during fluorescent microscopy without the need for additional labeling of microcapsules after histological fixation of tissues. The microcapsules were composed of in total 12 layers of oppositely charged polyelectrolytes: poly(allylamine hydrochloride) (#283215; Sigma-Aldrich, USA) and poly(sodium 4-styrenesulfonate) (#243051; Sigma-Aldrich, Belgium), and covered by the final 13th layer of poly(L-lysine)-graft-poly (ethylene glycol) co-polymer (#SZ34-67; SuSoS, Switzerland). The average size of PMs-PEG was determined using fluorescent microscopy as 2.7 ± 0.6 µm.

Animals and housing

Young, sexually mature zebrafish D. rerio (4–12 months old; body weight 0.3–0.5 g) were obtained from a local store. Fish of this age have fully formed organs and a mature immune system (Lam et al., 2004). Animals were placed in 4.5 L plastic bags with bottled water from Lake Baikal (Aqua LLC) with five individuals in each for transportation. The required water quality was maintained by constant aeration and daily replacement of 20% of water volume. The zebrafish were fed with commercial feed TetraMin once a day. The water temperature during the experiment was within the range of 21.9–22.6 °C, the pH was maintained at 7.34–7.86, and the oxygen level was 5–8 mg/L. The fish were acclimatized for one week in these conditions before the start of the experiment.

Injection of microcapsules into fish kidney

All experimental procedures were conducted in accordance with the EU Directive 2010/63/EU for animal experiments and the Declaration of Helsinki; the protocol of the study was registered and approved before the start of the experiment by the Animal Subjects Research Committee of the Institute of Biology at Irkutsk State University (Protocol No 1/2017).

To deliver the microcapsules into the fish bloodstream, we injected the microcapsules directly into the fish kidney, as described in detail elsewhere (Borvinskaya et al., 2018b). Due to the small size of D. rerio, it is difficult to administer PMs-PEG directly into the fish blood vessels; therefore, injection into vascular-rich tissues to deliver microcapsules into the circulatory system was performed as an alternative.

On the day of the experiment, the fish chosen from a random aquarium by lots were individually transferred to a solution of an antiseptic (0.0002% methylene blue for 1.5 min), then anesthetized in an emulsion of clove oil in water (0.1 ml/L) until the fish turns on its side and stops responding to a light pinch of the fin (about 1 min). Individuals were immobilized on a wet sponge and injected with 1.6 µl of microcapsules suspension in 0.9% NaCl (4.2*106 microcapsules per µl) into a central bulge of the fish trunk kidney using a 31G (Ø0.25 mm) needle connected to an IM-9B microinjector (Narishige, Japan).

To check the success of PMs-PEG delivery to the bloodstream, a rapid visual inspection of the gills with a fluorescent microscope was made after cutting the fish gill cover (Borvinskaya et al., 2018b). When the injection was performed correctly, fluorescent microcapsules occur in the capillaries of the denuded fish gills; we aimed to obtain several dozens of PMs-PEG per gill, which is enough to perform physiological measurements with a microencapsulated probe (Borvinskaya et al., 2017). In the control group, the fish received intrarenal injection of saline. Finally, fish were rinsed with water and returned to their holding bags. All staff conducting the experiment was aware of the allocation of the fish groups during the experiment. The fish condition and mortality were monitored twice a day.

Histological and histopathological examination

In order to obtain a sufficient number of samples for histological analysis, taking into account the expected mortality upon injections (Borvinskaya et al., 2018b), a total of 26 fish were injected. To study the possible long-term effects associated with microcapsule administration, on the day 1, 14, and 22 post-injection 3 individuals (n = 3) were chosen by lot from control and experimental groups as the minimum number of fish necessary to adjust the results for individual variability (in total 9 fish were inspected in both control and experimental groups). Fish were euthanized in a 0.5 ml/L clove oil-water suspension until the fish turns on its side and stops responding to a pinch of the fin. Since our studies included an assessment of the long-term effects of microcapsule administration on fish, including acute toxicity and mortality and organ pathologies, unscheduled animal euthanasia was not applied. The criterion established in advance to exclude the animal from the experiment (with subsequent treatment) was the presence of external signs of an infectious disease (grows and fouling or ulcers on the skin, progressive deterioration of the fins). Since no animals with external signs of infection were found, all animals from the experiment were included in the analysis.

For microscopic preparations, zebrafish were dissected in half sagittally and fixed in Bouin’s solution (Sigma-Aldrich, USA) at 22 °C for 1 day. Then, samples were washed for an hour in running tap water, and then stored in a solution of 10% buffered neutral formalin (Sigma-Aldrich, USA) at room temperature. Paraffin embedding was done using an STP 120 Spin Tissue Processor (Thermo Fisher Scientific, USA), according to the protocol for piscine tissues (Mikodina et al., 2009). Paraffin molds of dehydrated and paraffin infiltrated tissues were made using a MICROM paraffin embedding center EC-350 (Thermo Fisher Scientific, USA). Embedded tissues were sectioned at 6 µm in thickness with an HM-440 microtome (Thermo Fisher Scientific, USA). All fixed tissues were simultaneously subjected to histological processing operations to reduce the variability of the finished samples. Obtained slides were examined using a BX51 fluorescent microscope (Olympus, Japan) with a ×10 eyepiece and objective lenses of ×5, ×20, ×40, and ×60 magnification to detect microcapsules loaded with the fluorescent dye in fish tissues. Next, microsections were deparaffinized and stained with hematoxylin (Sigma-Aldrich, USA) and eosin (Sigma-Aldrich, USA) using staining containers line (BioOptica, Italy) with standard staining procedures. Obtained slides were examined by bright field microscopy using an Olympus BX51 microscope with a D7100 camera (Nikon, Japan) connected to the microscope. All photos were analyzed with ImageJ 1.50d (NIH) software (Schneider, Rasband & Eliceiri, 2012).

Data analysis

Histopathology was assessed blindly as histological slides were labeled with numbers coding for a sample description. Observed morphological alterations in the gills, liver, and kidneys of individual fish were recorded in a table of histological examination (Table S1) using a semiquantitative grading system (Gibson-Corley, Olivier & Meyerholz, 2013). Generally accepted markers of fish histopathology were selected for evaluation (Roberts, 2012; Mekkawy et al., 2013; Wolf et al., 2015; Wolf & Wheeler, 2018). In order to identify potential abnormalities, avoid misdiagnosis, and identify the presence of processing artifacts the organ morphology was also compared with published zebrafish specimens of the normal anatomy and histology (Menke et al., 2011; Cheng, 2004; McCampbell, Springer & Wingert, 2015).

Two indices were quantified for every type of morphological parameter on each examined histological section: affected area and prevalence (Table 1). For different types of morphological structures, such as organ parenchyma, circulatory system, bile canaliculi, renal glomeruli, renal tubules, gill lamellae and filaments, these parameters were evaluated independently of the rest of the tissue of the examined organ.

Table 1 Grading of the prevalence and affected area for each type of examined morphological parameters.

Points	Prevalence, number of separate cells or foci on section	Affected area, % of area with altered morphological structures (glomeruli, tubes, parenchyma, etc.) from the total area of these morphological structures	
1	1	<15	
2	2–3	15–50	
3	>3	>50	

Then mean recorded prevalence and affected area of viewed sections were multiplied on the proportion of sections with pathology to take into account how much tissue was affected in the third dimension. Thus, the equation for evaluation of final relative severity (S) of the particular histopathology for each individual was: S=M¯P×M¯A×npn

where

M¯P—mean grade of pathology prevalence

M¯A—mean grade of affected area

n—number of observed sections

np—number of observed sections with pathology

It should be noted that the evaluation of the severity and adversity of the morphological alterations, which is usually challenging (Schafer et al., 2018), was not a direct purpose of this study. Instead, we aimed to quantify the effect of microcapsules on the fish organism and directly compare the obtained grades for morphological structures in fish treated with PMs-PEG and injected with saline.

The obtained grades for individual morphological structures of the fish from the treatment groups were compared with the nonparametric Mann–Whitney test with Benjamini–Hochberg correction for multiple testing. Mulitple groups were compared using ANOVA with sex and post-injection day as factors. The analysis was performed with the coin (Hothorn et al., 2008), magrittr (Schmidt, 2019) and base stats packages for R (R Core Team, 2017).

Results and Discussion

Fish mortality and distribution of PMs-PEG in fish organs

Fluorescent PMs-PEG were implanted into the circulatory system of adult zebrafish D. rerio by intrarenal injection. The highest mortality (5–6% in both experimental and control groups) occurred immediately after the injection due to the stress and injury caused by the injection. Those fish that woke up after injection with saline did not die during the experiment. The cumulative mortality of fish that received an intrarenal injection of PMs-PEG during the first week was 18%. Further presence of microcapsules in the fish did not cause noticeable effects at the organismal level, thus suggesting a low level of direct toxicity of PMs-PEG. This outcome is consistent with those obtained earlier for fish received an intrarenal injection of 1.6–5 µm diameter PMs-PEG (Borvinskaya et al., 2018a).

The microstructure of the kidneys, liver, and gills of fish was monitored for 22 days after the injections to clarify interaction of PMs-PEG with fish tissues. During an injection into the fish kidney, most of the microcapsules spill into the abdominal cavity. Therefore, the largest number of microcapsules on histological sections can be seen around the swimming bladder, at the bottom of the abdominal cavity and between the internal organs. Moreover, a large concentration of microcapsules is present in the renal parenchyma at the injection site.

The microcapsules that had entered blood vessels (due to their rupture with a needle) were detected in the peripheral parts of the kidney one day after injection, as well as in the liver and gills (Fig. 1, Table S2). First day observations show that only a minimum amount of microcapsules entered the skeletal muscles and brains of the fish. These histological observations are generally consistent with the previous external examinations of fish organs after the PMs-PEG intrarenal administration, which demonstrated immediate distribution of microcapsules mainly in the liver, gills, kidney, and less often in the muscles, gonads, and fin capillaries (Borvinskaya et al., 2018a).

Two weeks after injection, histological examination revealed that the microcapsules were still present in the kidneys and liver of the fish, supporting the observation of PMs-PEG accumulation in organs with a rich capillary network. In the second week after injection, microcapsules also appeared in the intestinal epithelium and single PMs-PEG were found in the spinal cord and eyes. In the muscles and brain, the microcapsules were still located, albeit rarely and singly.

In the third week, the microcapsules were still common in the kidneys and liver, and were also abundant in the gills, intestine and muscles of fish (Fig. 1). Separate microcapsules and aggregates were found in skeletal muscles, in the brain and spinal canal. Except for one individual with plenty of PMs-PEGs observed in the heart (Fig. 2A), in other fish only rare microcapsules were found in central vessels (Fig. 2).

We did not observe an increase in the number of PMs-PEGs in the liver, kidneys and gills of fish during the three weeks after injection (Table S2), suggesting that the majority of microcapsules carried with the blood throughout the fish body quickly settles in these organs as what has been previously shown in mouse models (Sindeeva et al., 2020). However, at least some PMs-PEG were able to migrate from the tissues where they were originally settled and accumulated in the skeletal muscles and intestine.

Figure 1 Distribution of PMs-PEG in zebrafish organs after intrarenal injection.

(A–C) Schematic of PMs-PEG distribution three weeks after injection (mean number of microcapsules per histological section averaged for three individuals). (D) Fluorescent image of PMs-PEG (arrows) in hepatic parenchyma. (E) Fluorescent image of PMs-PEG (arrows) in gill lamellae. (F) Fluorescent image of PMs-PEG (arrows) in renal parenchyma and posterior cardinal vein (v). M, skeletal muscles; Br, brain; K, renal parenchyma; L, liver parenchyma; G, gill lamellae; H, heart; E, eyes; I, intestinal wall.

Figure 2 PMs-PEG in zebrafish circulatory system after intrarenal injection.

(A) Fluorescent image of PMs-PEG in the hepatic vein (arrowhead) 14 days after injection. (B) Fluorescent image of PMs-PEG in posterior cardinal vein (arrowhead) of the fish kidney (k) 14 days after injection.

The histological examination of the D. rerio intestine showed that engulfed PMs-PEG were carried into the intestinal mucosa by phagocytic cells (Fig. 3). These loaded phagocytes tend to gather in the intestinal epithelium, forming cavities between enterocytes, which can rupture (Fig. 3D) and release their contents into the intestinal lumen (Fig. 3C). Thus, fluorescent particles enter the intestinal lumen (Figs. 3C, 3D) and can be excreted in feces. This indicates that the fish can at least partially get rid of the irritating agent implanted in the body. However, the existence and effectiveness of such a mechanism of microparticle excretion have almost never been studied.

Figure 3 PMs-PEG in zebrafish intestine after intrarenal injection.

(A) Fluorescent image of intestine section shows microcapsules located in the intestinal epithelium (arrows) 22 days after injection. (B) Intestinal epithelium with phagocytes that engulfed non-biodegradable PMs-PEG passing through the intestinal wall (arrows), H&E stain. (C) Fluorescent image of PMs-PEG (arrows) appeared in the intestinal lumen between feces (f) and intestinal wall (i) 14 days after injection. (D) Enlarged image of region from C after H&E stain of histological section showing phagocytes loaded with PMs-PEG (arrows) in the intestinal epithelium (e). The arrowhead points to a phagocyte that spills out into the intestinal lumen and releases microcapsules, after which an empty cavity (asterisk) and rupture remain in the intestinal epithelium.

It is known that particles larger than 0.5 µm cannot pass through the epithelium of blood vessels and thus cannot be removed through the excretory organs (De Jong et al., 2008). Therefore, they usually remain in the body and cause chronic inflammation, which at best results in their covering with connective tissue and encapsulation. In available literature only the internalization in the intestine, but not excretion of particles larger than 1 µm has been described in vertebrates (Reineke et al., 2013; Lu et al., 2016; Deng et al., 2017; Wang et al., 2019; Avio, Gorbi & Regoli, 2015). In these studies, in which animals received microparticles through the gastrointestinal tract, it is impossible to determine whether all microparticles in the intestinal epithelium were in the process of internalization or some of them were excreted from the body. In turn, in studies of the biodistribution of microparticles injected intravenously or intraperitoneally, researchers, as a rule, focus on other organs, but not the intestine (Kaminski et al., 2004; Decuzzi et al., 2010), or available studies are not detailed enough (Bartholdson et al., 1977; Westrøm et al., 2018). Therefore, further research could clarify the specificity and effectiveness of observed migration of loaded phagocytes, followed by penetration of the mucosal epithelium and microparticles removal from D. rerio intestine. Nevertheless, the acute inflammation caused by phagocytes loaded with PMs-PEG likely promotes rupture and desquamation of the intestinal epithelium and is quite damaging for the organism.

It is not entirely clear from examined histological sections whether phagocytes were brought into the D. rerio intestine with blood or migrated through the intestinal wall from the nearest regions of the body cavity where a large number of microcapsules were spilled. Previously, long-distance transport of PMs by phagocytic cells through the body has been shown in mice and zebrafish (De Koker et al., 2010; Borvinskaya et al., 2018a). In mammals, phagocytes that have engulfed the antigen migrate deep into the body to the lymph nodes. In fish there are no lymph nodes (Rummer et al., 2014), but activated phagocytes can migrate from inside the body to the skin of the fish (Borvinskaya et al., 2018a). Therefore, the recorded delay between PMs-PEG administration and accumulation in fish intestine could be caused by the slow migration of loaded phagocytes through the fish tissues.

Progression of immune response

For non-biodegradable microcapsules it is important to evaluate the long-term effects of their introduction into various organs of fish. For this purpose, samples collected 1 day and 2–3 weeks after injection were analyzed. The immune response fully develops by the day after the traumatic effect (Antonio et al., 2015), so at this time it is possible to assess the general state of the fish immune system. Between the second and third weeks after the injury, the acute phase of inflammation ends, associated directly with the removal of damaged tissue structures, and the regeneration process begins (Wahli et al., 2003). Therefore, if the immune response progresses during this period, this indicates that immunogenic material is present in the body.

On the first day after the administration of PMs-PEG, the vast majority of microcapsules in D. rerio sections were found spilled in the body cavity of the fish. Most microcapsules laid freely between internal organs, but some of them were agglomerated inside large phagocytosing cells with pale eosinophilic cytoplasm and a large basophilic nucleus. The phagocytosis of PMs-PEG was observed outside the injection site and associated blood clot; thus, it did not result from non-specific phagocytosis of cell debris or infectious agents in the wound. This indicates that despite the polyethylene glycol coating, PMs-PEGs were recognized by immune cells shortly after injection. The obtained results are consistent with in vitro tests, which suggest that mammalian phagocytes internalize a portion of microparticles coated with polyethylene glycol-containing polymer within 4 h (Wattendorf et al., 2008). Two weeks later all microcapsules in the body cavity were engulfed by phagocytes, which became “foamed” because of a large number of microparticles (Figs. 4A, 4B).

Figure 4 Macrophage phagocytosis of PMs-PEG.

(A) Body cavity of D. rerio 14 days after the PMs-PEG injection into the fish kidney with the microcapsules (transparent vesicles) collected inside the phagocytes. (B) An enlarged image of D. rerio phagocytes with engulfed PMs-PEG (arrowheads). (C) Aglomerates of phagocytes with engulfed PMs-PEG (arrowheads) between the liver (l) and spleen (s) 14 days after PMs PEG injection in fish kidney. (D) Enlarged image of region from C shows a focus of inflammation in hepatic parenchyma. There are many phagocytes with engulfed microcapsules (arrow) and eosinophilic granular cells (arrowhead).

The recognition of the implant by the immune system leads to chronic inflammation in the wound, which lasts until the foreign body is destroyed or isolated (Anderson, Rodriguez & Chang, 2008). To investigate the progression of the immune response provoked by PMs-PEG, we examined the tissue microenvironment of the microcapsules stuck in different fish organs. In one of the fish injected with PMs-PEGs, an abundance of microcapsules spilled into the body cavity and caused considerable inflammation to the surface of the spleen and liver on the 14th day after injection (Figs. 4C, 4D). The inflamed spleen increased in size and fused with the adjacent edge of the liver. There were many microcapsules at the place of their confluence causing further inflammation. In the liver close to the sites of PMs-PEG agglomeration, the parenchyma became loose due to its replacement with phagocytic cells and eosinophilic granular cells (mast cells) (Fig. 4D), indicating severe parenchyma dysfunction at the inflamed organ edge. Similar immune reactions were observed in the kidneys two and three weeks after injection around the implanted fluorescent material (Fig. 5C). There were many leucocytic cells surrounding the fluorescent microparticles; however, given the hematopoietic function of the fish kidney, it has not been established whether these cells represent normal renal parenchyma or were recruited due to inflammation. Both in the liver and in the kidney, in the rest of each organ (i.e., free of microcapsules) hepatic and renal parenchyma looked intact, with no signs of degeneration. This indicates that the irritant effect of PMs-PEG occurred locally.

Figure 5 Granulomas in D. rerio injected with PMs-PEG into the kidney.

(A) Fluorescent image of a granuloma (arrow) located in the abdominal cavity of the fish between liver (l) and ovary (o) 14 days after PMs-PEG injection. (B) An enlarged image of region from A shows giant cell surrounded by concentric layers of macrophages loaded with microcapsules (arrow). (C) Site of immune cells attraction in renal parenchyma 14 days after PMs-PEG injection in fish kidney. There are many phagocytes with engulfed microcapsules inside (arrow) and eosinophilic granular cells (arrowhead). (D) Granuloma (arrow) in the renal parenchyma of zebrafish 14 days after PMs-PEG injection.

Dense nodules with fluorescent material consisting of several layers of phagocytic cells and fibroblasts were found two weeks after PMs-PEG administration in the kidney and abdominal cavity of the zebrafish (Figs. 5A, 5B, 5D). These specific morphological structures were granulomas, formed as a result of attempts of immune cells to isolate the foreign bodies from normal tissues (Anderson, Rodriguez & Chang, 2008). The observed macrophage fusion around the implant with its subsequent coating with a fibrin capsule is the final stage of the immune system response to the irritant and occurs if the foreign body cannot be split and removed from the tissue. Therefore, the immune response to PMs-PEG delivered to fish organs through the circulatory system was shown to develop according to the classical foreign body reaction scenario. The same outcome of the healing process was observed previously in zebrafish muscles 22 days after intramuscular injection of PMs-PEG (Borvinskaya et al., 2018a). Despite loaded phagocytes being shown to carry microcapsules out of the implantation site (Borvinskaya et al., 2018a), the agglomerates of many microparticles were too large to be removed, and after about 2-3 weeks the implant was seen to be granulated.

On the fish gill sections, the microcapsules were found only as separate fluorescent objects in blood capillaries, never as agglomerates. Unfortunately, it was difficult to determine whether they were engulfed by phagocytes, as a result of the destruction of blood cells by Buen’s fixative. However, no signs of local inflammation were observed around them in gill lamelae.

Comparative morphology of fish kidney

As shown above, PMs-PEGs are immunogenic, therefore they can potentially cause not only local inflammation, but overstimulation of the immune system with severe toxic effects on the whole organism. In a hyperinflammatory state, tissue damage, functional organ failure and, ultimately, death can occur as a result of an uncontrolled cascade of inflammatory reactions (Philip et al., 2017). To assess the potential toxicity of microcapsules in vivo, we compared the morphology of organs of D. rerio that received an injection of either PMs-PEG or saline (Table S1).

The D. rerio kidney is an abdominal organ located along the spine of zebrafish and consists of tissues with mixed hematopoietic, immunologic, endocrine, and urinary functions (Wolf et al., 2015). The renal tissue of examined fish was represented by parenchyma from undifferentiated cells with condensed basophilic nuclei. Throughout the entire length of the parenchyma, there were numerous distal and proximal renal tubules and filtering renal glomeruli (renal corpuscles) (Fig. 6A).

Figure 6 Morphological structure of zebrafish kidneys after injection of saline into fish kidney.

(A) Normal kidney structure 22 days after the injection. The white arrow marks hyaline droplets in the cytoplasm of epithelial cells in proximal tubule; arrowheads mark lymphocytes crossing the epithelium of the tubules. (B) Melanomacrophage (arrow) in the parenchyma of a fish kidney 22 days after injection of saline into the kidney. (C) Granular material in the lumen of the proximal tubule (arrowheads) in the wound zone 1 day after injection. (D) Vacuolar degeneration of distal tubule (arrow) 22 days after injection. (E) Nephron neogenesis (arrow) in the kidney of a zebrafish 14 days after injection of saline into the kidney. (F) Neoplastic alterations found in the kidney of D. rerio 14 days after injection of saline with proximal tubule apoptosis (open arrowhead), thickening of the capsule of the renal glomerulus (arrowhead). (G) Enlarged image of region from F showing macrophages (black arrows) in the renal parenchyma. (H) Enlarged image of region from F showing eosinophilic granule cells (black arrows) in the renal parenchyma. Ds, distal tubule; P, proximal tubule; G, renal glomerulus; Er, red blood cells.

Injection of PMs-PEG or saline was performed into the central bulk of the fish trunk kidney. At the injection site, the tissues were mechanically damaged, and over the next three weeks, successive stages of the healing process were observed there. Where the renal artery was ruptured, blood spilled into the body cavity and formed a blood clot, which was gradually disassembled by relatively rare phagocytic cells that absorb fragments of blood cells. The hematoma in the abdominal cavity remained not resolved completely for three weeks after the injection. Hemosiderin-laden macrophages, large cells with a brown cytoplasm and a basophilic nucleus, were found in the renal parenchyma of fish from the control group on the first and 22nd day after injection (Fig. 6B). Melanomacrophages are normally present in kidneys of teleosts (Steinel & Bolnick, 2017) and are additionally recruited to the site of injury during the healing processes in the wound. Throughout the entire observation period, foci of newly formed nephrons (dark small basophilic nodules) were found in different parts of the kidney of the fish from the control group (Fig. 6E). This indicates the beginning of a regenerative process (nephron neogenesis), typical for adult fish (Wolf et al., 2015) and, probably, stimulated by tissue damage during injection.

In the fish that received saline, the renal parenchyma outside the injection site consisted of loose hematopoietic tissue containing a heterogeneous cell population of erythrocytic, thrombocytic, leukocytic precursors, and mature leukocytes (Fig. 6A). Blood cells were abundant in the sinuses of the renal parenchyma, which is typical for this hematopoietic tissue. Renal glomeruli in fish from the reference group were presented by a tuft of blood capillaries enclosed in a capsule of single-layered squamous epithelium (Figs. 6A, 6B). The distance between the glomerulus and the capsule wall (an indirect indicator of Bowman’s space) varied significantly (Table S1). In the same individual, it could be dilated (>1/3 of the diameter of the capsule) or be absent; thus, this diagnostic indicator was recognized by us as uninformative. The renal tubules in the kidneys were represented by tubular structures from one layer of columnar epithelial cells with large basophilic nuclei. The proximal tubule epithelial cells contained an eosinophilic cytoplasm and periodically had a moderate accumulation of hyaline droplets (Fig. 6A), which in teleosts is not associated with glomerular damage (Roberts, 2012). Furthermore, there were small lymphocytes with highly condensed basophilic nuclei in some spots between the mature epithelial cells of the proximal tubule (Fig. 6A). The epithelial cells of the distal tubules had nuclei displaced to the basement membrane and a pale strongly vacuolated cytoplasm (Fig. 6A), which is normal for D. rerio (McCampbell, Springer & Wingert, 2015).

Such pathologies as single nephron degeneration and vacuolar degeneration (Fig. 6D) of the distal tubule were sporadically found in fish from the control group. The clots of organic material were also periodically detected in the lumen of the renal tubules of fish that received saline. At the injection site, the lumen of the tubule was filled with granular material, probably fragments of destroyed cells (Fig. 6C). In one individual from the control group, a neoplastic process associated with massive lesion of renal tissue was found in the head kidney on day 14 after injection (Fig. 6F). In a rather large field, the renal parenchyma was replaced by spindle anaplastic cells with eosinophilic cytoplasm and large basophilic nuclei. Degeneration and extensive necrosis of the renal tubules and glomeruli, together with massive involvement of phagocytes and eosinophilic granule cells, indicated ongoing acute inflammation in the affected area.

In the kidney of fish that received PMs-PEG, the injection site was a lot of shapeless loose material, consisting of coagulated red blood cells mixed with microcapsules. Two and three weeks after injection, severe inflammation and foreign body reaction were observed around the fluorescent material, as described above (Figs. 5C, 5D). In the regions remote from the injection site the normal architecture of trunk kidney was generally maintained: the distal and proximal renal tubules and renal glomeruli were clearly distinguishable; the parenchyma between them was represented by hematopoietic tissue, consisting of immature lymphocytes, as well as leukocyte precursors and erythrocytes (Fig. 7A).

Fish from the control group and fish treated with PMs-PEG statistically did not differ in the level of estimated morphological parameters. On the second week after injection, in the renal parenchyma and around the glomeruli of fish that received microcapsules, the number of pigmented macrophages was higher than in fish of the control group (Fig. 5C), which indicates an increase in the inflammatory process. Macrophages directly absorbed microcapsules and were involved in the absorption of damaged erythrocytes or nephrons. However, on the third week after injection, the opposite effect was observed: the number of macrophages found in the renal parenchyma was even lower in fish from the experimental group than in fish from the control group. This probably indicates that three weeks after the injection, the generalized inflammation ceased, while the foci of inflammation remained directly around the aggregates of the microcapsules.

Figure 7 Morphological structure of zebrafish kidneys after injection of PMs-PEG into fish kidney.

(A) Zebrafish kidney 14 days after the injection. The arrow marks a swollen renal glomerulus. (B) Thickening of renal glomerulus capsule (arrow) and melanomacrophage (arrowhead) in the kidney of D. rerio 22 days after injection of PMs-PEG. (C) Proximal tubule degeneration with number of lymphocytes (arrows) and eosinophilic granule cells (arrowhead) crossing the epithelium of the tubules 22 days after injection of PMs-PEG into the kidney. (D) Proximal tubules necrosis (arrow) in fish kidney 22 days after PMs-PEG injection.

On the second and third week after capsule administration foci of early stages of renal glomerular fibrosis (Fig. 7B) and dilated capillaries in the renal glomerulus (Fig. 7A) were observed in the wound of the site of injection. Foci of intense necrosis of individual renal tubules were also found in fish from the experimental group on the 22nd day after injection (Figs. 7C, 7D). The severity of these conditions can be considered mild since they were sporadic and did not affect the adjacent renal tissues, which retained their ability to function normally.

Such morphological pathologies such as blood clots, vascular fibrosis, renal glomerular necrosis, and fibrosis, as well as granuloma formation, were not found outside of the injection site. The number of new immature nephrons, a diagnostic indicator of the regeneration process, did not differ significantly in the fish kidneys of either the control or the experimental group. Therefore, the inflammatory process in the response of D. rerio kidney to the PMs-PEG administration can be characterized as moderate and local.

Comparative morphology of fish gills

Normal morphology of gills was observed in the fish from both control and experimental groups (Fig. 8A). The organ consisted of cartilaginous gill arches with outgrowths of gill filaments with two rows of secondary lamellae. Between lamellae, filaments were coated with 1–3 rows of squamous pavement cells. No disturbances in the structure of filaments, such as massive hyperplasia of filament epithelium, fusion of filaments, fibrinolysis, and leukocyte infiltration, were found. There were also no vasculature disorders, with the exception of several cases of hyperemia of gill filaments. Epithelial folds that were secondary lamellae were well separated from each other and in cross-section looked like a thin channel formed from 1–2 layers of squamous cells and filled with red blood cells lined up in a row. In fish from the control group some small zones of lamellar epithelial hyperplasia were noted. On the 14th day after injection, abundant extracellular material rich in eosinophilic protein, presumably mucus (Saleh et al., 2018; Sveen et al., 2019), was found on the surface of the lamellae in two reference fish (Fig. 8B). This gill covering is likely to perform a protective function, reducing the availability of the gill epithelium to irritating agents from the external environment, and its secretion by epithelial cells may be an artifact associated with the use of clove oil as an anesthetic. However, no severe lesions of the lamella structure as lifting or desquamation of the respiratory epithelium, aneurysms, and other circulatory disorders were found in fish that received an injection of saline.

Figure 8 Morphological structure of the gills of zebrafish.

(A) The photo of the gill cavity of the zebrafish 14 days after the injection of saline depict gills with normal architecture. (B) The gills of the fish covered with extracellular eosinophilic material (arrowhead) 14 days after the injection of saline. (C, D) Hyperplasia of the respiratory epithelium (arrowhead) and partial fusion of the gill lamellae (open arrowhead) in the gills of fish 22 days after PMs-PEG injection. F, gill filament; L, gill lamellae; Er, red blood cells.

A comparative analysis of the gill microstructure after saline or PMs-PEG introduction into the kidney did not reveal significant differences in the frequency or area of the studied histopathologies. This indicates a relatively small effect of systemically administered microcapsules on the structure and function of this organ. A rather mild violation of the structure of gill tissue, such as the fusion of individual secondary lamellae, was detected in all fish three weeks after injection of PMs-PEG. This was either direct adhesion of neighboring lamellae or filling of interlamellar sulci due to epithelial hyperplasia. As hyperplasia lasted only for a couple/several neighboring lamellae, without forming continuous fusion, therefore the lesion area was too small to seriously decrease in the surface of gas exchange and impair respiration. Lamella fusion is often a non-specific response to chronic inflammation that causes proliferation of a mixed population of the pavement, mucous and chloride cells and/or leukocyte infiltration (Wolf et al., 2015). In PMs-PEG-injected fish, sporadic respiratory epithelial hyperplasia may be caused by a general increase in immunoreactivity due to the irritating effect of the microcapsules.

Two and three weeks after PMs-PEG injection, a slight increase in the frequency of lifting and desquamation of the epithelium (Table S1) that line the blood capillaries was recorded. Epithelial lifting is the initial stage preceding the rupture and loss of the epithelium, which is considered as a severe pathology since it directly violates the respiratory process. In fish injected with microcapsules, such changes were sporadic and could not significantly affect the respiration of the fish, especially considering regenerative capabilities of gill tissue (Wolf et al., 2015). Also, many authors report that such violations are often artifacts, a consequence of mechanical damage to tissue during the cutting of histological sections (Frasca et al., 2018; Wolf et al., 2015).

Since the gills are intensively supplied with blood, one of the expected adverse effects from the systemic administration of a large number of PMs-PEG was the agglomeration of microcapsules in the finest gill capillaries. Formation of blood congestions followed by severe anoxia can cause direct toxicity of the microimplants for the fish. Within three weeks after microcapsule administration, we revealed no signs of circulation obstructions (aneurysms, blood clots, etc.) in primary and secondary gill filaments. The elastic polymer shell of the PMs, which allows them to squeeze through narrow spaces, probably contributes to their satisfactory hydrodynamic characteristics in fish capillaries.

Comparative morphology of fish liver

The hepatic structure of fish from the control group, which received an injection of saline into the kidney, was typical for zebrafish (Cheng, 2004). We observed homogeneous parenchyma consisted of dense, uniform, angularly shaped hepatocytes with soft margins with a large basophilic nucleus in the center and a very vacuolated, clear cytoplasm (Fig. 9A). Large vacuoles accumulating glycogen have been reported to be typical for zebrafish hepatocytes, especially for captive fish fed with commercial feed (Wolf, Wheeler, 2018). There occasionally were small basophilic stained macrophages (Kupffer cells) squeezing between hepatocytes (Figs. 9A, 9C). Within the hepatic parenchyma, there were many sinuses and blood vessels moderately filled with red blood cells and bile canaliculi lined with basophilic epithelial cells. The presence of hepatocytes with pyknotic nuclei was detected in many fishes from the control and experimental group (Fig. 9B), which may be an artifact that occurred during the fixation of specimens (Wolf, Wheeler, 2018).

Figure 9 Morphological structure of zebrafish liver after PMs-PEG injection into fish kidney.

(A) Photo of zebrafish liver 22 days after the injection of PMs-PEG depict normal hepatic parenchyma (P) with hepatic vein (arrowhead) and an oblique cut of the bile duct (open arrowhead). Arrows indicate hepatic macrophages. (B) Lymphocytic infiltration (arrow) near the bile duct (arrowhead) 22 days after administration. An open arrow indicates pyknotic nuclei. (C) Eosinophilic granule cells (arrow) under the connective tissue surrounding the liver, 14 days after injection. Arrows indicate hepatic macrophages. (D) Eosinophilic granule cells (arrows) at higher magnification in the hepatic blood vessel.

No statistically significant differences in hepatic morphology were revealed when comparing the microstructure of the liver of fish that were injected with PMs-PEG or saline. Similarly, as was described for other tissues, the immune response to the implant was local, and the liver structure in the surrounding space was without histopathologies. Generally, microcapsules were separately located in the fish liver and were surrounded by normal parenchyma. In one individual, a large number of PMs-PEG forming agglomerates were stuck on the liver surface (Figs. 4C, 4D) and caused a severe inflammation and damage of the hepatic parenchyma near the edge of the liver 2 weeks after the PMs-PEG injection (see above). PMs-PEG-injected fish also showed an increase number of eosinophilic granule cells (mast cells), which are often found in chronically inflamed tissues (Reite & Evensen, 2006). Eosinophilic granule cells in zebrafish liver were typically localized under connective tissue surrounding the liver or around hepatic vessels and bile canaliculi and rare in the hepatic parenchyma (Figs. 9C, 9D).

The bile canaliculi architecture in the liver of the PMs-PEG injected fish was normal in all the studied samples. Several small accumulations of leukocytes around the bile canaliculi were detected in the liver of the PMs-PEG injected fish (Fig. 9B); however, no pronounced inflammation, proliferation of connective tissue or necrosis were detected around them. There were no inclusions or other material in canaliculus lumen and no bile pigment in the cytoplasm of hepatocytes, confirming the normal bile canaliculus functioning.

Conclusions

The study gives a general histopathological assessment of short-term and long-term effects of non-biodegradable PMs-PEG administration on the state of the D. rerio organism. With intrarenal administration, the microcapsules did not accumulate in organs outside of the injection site for weeks, so those of them that entered the bloodstream seemed to settle rapidly in the blood vessels. Therefore, the most pronounced immunogenic effect of the microcapsules was in the part of the kidney into which they were injected. In those sites where a large number of microcapsules accumulate, severe inflammation associated with tissue lesions occurred in the first weeks after administration. This is relevant for the fish intestine, but in this case, microcapsules engulfed by macrophages could be brought from adjacent body cavity rather than the bloodstream.

Kidney damage and spilling of numerous PMs-PEG to the kidney parenchyma and abdominal cavity were caused by the intrarenal injection approach for microcapsule introduction into the zebrafish bloodstream, which was the main limitation of this study. At the same time, the number of fluorescent microcapsules observed in fish gills right after injection was close to the minimal amount required for measurements of blood parameters (Borvinskaya et al., 2018b). Therefore, the use of a more precise delivery technique allowing the introduction of a much smaller amount of PMs-PEG but exclusively into bloodstream could significantly reduce the observed adverse effects of the microcapsules.

This is confirmed by the fact that separate capsules brought by blood to the distal organs did not cause a noticeable effect on the surrounding tissue. The morphology of gills, liver and in those parts of the kidney that were remote from the injection site was not significantly altered in treated fish compared to the fish injected with saline. Such markers of generalized inflammatory reaction as leukocyte infiltration, extensive necrosis, extensive perivascular and periductal fibrosis and scar tissue formation, bile obstruction etc. were absent in the fish organs. PMs-PEG in the bloodstream did not cause blood conjunction, thus demonstrating a low aggregation capacity. In general, administration of PMs-PEG to blood does not have a systemic effect on fish health, which remains satisfactory.

Therefore, it is reasonable to use non-biodegradable PMs for the systemic introduction of toxic probes, for which general and local toxicity is greater than the local effect of the immune response to the microcapsules. Considering that PMs-PEG can be engulfed by phagocytes, long-term measurements (lasting a day or more) using chemical sensors loaded into non-biodegradable microcapsules are meaningful only for molecules diffusing across the cell membranes. Importantly, long-term studies should be carried out with caution both due to possible alterations of environment surrounding PMs caused by developing inflammation process and to possible migration of phagocytes loaded with PMs from the site of their initial deposition or even its partial excretion from the body. The results obtained on the fate of microcapsules in the circulatory system and organs of fish can be attributed to other vertebrates.

Supplemental Information

Table S1 Relative severity (S) of diagnostic parameters of zebrafish pathology calculated from mean grades of pathology prevalence, affected area and number of observed sections

Click here for additional data file.

Table S2 Distribution of PMs-PEG in fish organs (average number of microcapsules per histological section)

Click here for additional data file.

Supplemental Information 1 ARRIVE Author Checklist

Click here for additional data file.

The research was partially carried out using the equipment of the Core Facility of the Karelian Research Centre of the Russian Academy of Sciences. The authors are grateful to Dr. Sveltana Murzina (Institute of Biology of the Karelian Research Centre of the Russian Academy of Sciences) for training in histological technique, critically reading the manuscript and histoadvising.

Additional Information and Declarations

Competing Interests

Author Contributions

Animal Ethics

Data Availability

The authors declare there are no competing interests.

Ekaterina Borvinskaya conceived and designed the experiments, performed the experiments, analyzed the data, prepared figures and/or tables, authored or reviewed drafts of the paper, and approved the final draft.

Anton Gurkov conceived and designed the experiments, analyzed the data, authored or reviewed drafts of the paper, and approved the final draft.

Ekaterina Shchapova and Andrei Mutin conceived and designed the experiments, performed the experiments, prepared figures and/or tables, and approved the final draft.

Maxim Timofeyev conceived and designed the experiments, authored or reviewed drafts of the paper, and approved the final draft.

The following information was supplied relating to ethical approvals (i.e., approving body and any reference numbers):

Animal Subjects Research Committee of Institute of Biology at Irkutsk State University provided full approval for this recearch (Protocol No. 1/2017).

The following information was supplied regarding data availability:

Histological images supporting Table S1 are available in figshare:

Borvinskaya, Ekaterina (2021): Histological photos of D. rerio tissues taken after 1, 14 or 22 days after the injection into fish kidney of non-biodegradable polyelectrolyte microcapsules coated with polyethylene glycol (PMs-PEG) or saline. figshare. Figure. https://doi.org/10.6084/m9.figshare.13729069.

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
