# Peer review of "Histopathological analysis of zebrafish after introduction of non-biodegradable polyelectrolyte microcapsules into the circulatory system"

_PeerJ, doi:10.7717/peerj.11337_

## Round 0.1 · original submission · Major Revisions

The conclusion from 3 reviewers is that your manuscript needs a revision. Please, respond to all comments point by point. I am waiting for your revised version.

Reviewer 1 ·

Basic reporting

The manuscript “Histopathological analysis of zebrafish after introduction of non-biodegradable polyelectrolyte microcapsules into the circulatory system”

The paper contains information of potential interest considering the relevance of the histolopathology of zebrafish species after introduction of non-biodegradable polyelectrolyte microcapsules. However, some critical points still need to be addressed

Experimental design

Abstract:

- The histopathological data in the abstract is a bit confusing! I suggest that histopathological data in several organs of zebrafish as brief sentences should be included.
- Could author(s) please say something about the benefit of this study?


Introduction:
- It is well-written and showed the good introduction; however, Lines 44-48, 51-52, 91-95; some relevant references to quote these sentences should be provided.


Materials and methods:

- Line 115: I am confusing about why the age of zeabrafish, as 4-12 month!
- Line 152: why did author(s) collect data the day 1, 14 and 22 post-injection?
- Regarding the fluorescent protocol, I recommend separating this method into new paragraph! provide information about the antibodies, provide details on the antigens (native, recombinant or peptides) and specificity of the antibodies. Refer to previous publications that validated and used this antibody.

Validity of the findings

Results and Discussion:
- Please include the figure (s) in Lines 271-273.
-
- Please add the figure in Line 275 (after.., while they were abundant in the sinuses in the organs). Additionally, they … refer to?
-
- Line 294-296: I would recommend adding some discussion to support!!! like literature reviews or scientific reports.

- Many figures displayed as Fig. 3C-3D (Line 299), Fig. 4A,4B (Line 332) or Fig. 9A, 9D (Line 573). All words “Fig.” should be replaced with “Figs.” instead.

- What does it mean about … with dissociation and atrophy of hepatocyte ? (Lines 358-359)
- Is this statement correct? I did not see the figure in the abdominal cavity (Line 369).

- Please add some relevant references in Lines 399-401.
-
- Figure 3: What does “f” in Figure 3D stand for? Why do you think that the release of microcapsules is exhibited? Is it artifact?

- Figure 5: Better replace ‘oocyte’ in Figure 5A with ‘ovary’. What do you mean that it is the focus of inflammation in Figure 5C? Could you please explain in more detail?

- Figure 6: It is not easy to identify the macrophage phagocytizing renal cell and mast cell !!. please add the higher magnification of the 6F. Importantly, I recommend that the mast cell should be replaced by the eosinophil granulocyte instead.

- Figure 7: Could you please indicate the dilated capillaries in this figure?

- Figure 8: What is the mucus in Figure 8B?. I guess that the mucus cell is shown; however, I think that this is the fiber of connective. Moreover, Figs. 8C-8D, they are the lamellar fusion together with epithelial hyperplasia.

- Figures 9: Accordingly, two figures displayed the mast cell; however, I recommend that they should be considered as eosinophil granulocytes as mentioned above. These cells had a diameter of 9-10 µm and large eosinophilic granules in the cytoplasm. Could you please show in higher magnification of this cell?

Reviewer 2 ·

Basic reporting

The manuscript needs recheck due to linguistic errors.
The gap of knowledge should be explained carefully.

Experimental design

Ethical code should be provided.

Validity of the findings

Study limitations should be provided.

Reviewer 3 ·

Basic reporting

It is quite clear statement of the rationale for your approach to the problem studied.

Experimental design

Fluorescent PMs-PEG detection is quite specific technique. The author(s) may set this method as a subsection and all details concerning antigens and antibodies used in this examination should be given.

Validity of the findings

-“Fig” or “Figs” Could the author(s) recheck throughout the paper?
-Lines 294-296 and 399-401, could author(s) should quote these sentences?
-What dose “f” in figure 3D stand for?
-Few words are ambiguous or incorrect. Could author(s) recheck them?
Like “oocyte” in figure 5A (ovary?) or “mast cell” in figure 6F and figure 9 (eosinophil granulocyte?)
-Some descriptions are missing or not related to the figures, Like…abdominal cavity in figure 5 and …dilated capillaries in figure 7. Could the author(s) point them out?

Additional comments

This paper is quite clear and understandable. The researcher conveys research in proper and concise manner. However, some valuable data should be provided and amended before publication.

Annotated reviews are not available for download in order to protect the identity of reviewers who chose to remain anonymous.

---

## Round 0.2 · accepted · Accept

Congratulations on your publication.

Reviewer 1 ·

Basic reporting

Clear observations are found throughout the manuscript.

Experimental design

This part in the paper is clear.

Validity of the findings

All data are satisfied for publishing.

Additional comments

All of the comments have been addressed and revised. The data provided in the paper is satisfied for publication.

Reviewer 3 ·

Basic reporting

-

Experimental design

-

Validity of the findings

-

Additional comments

-